# Association of Self-Reported and Device-Measured Sedentary Behaviour and Physical Activity with Health-Related Quality of Life among European Older Adults

**DOI:** 10.3390/ijerph182413252

**Published:** 2021-12-16

**Authors:** Oriol Sansano-Nadal, Maria Giné-Garriga, Beatriz Rodríguez-Roca, Myriam Guerra-Balic, Kelly Ferri, Jason J. Wilson, Paolo Caserotti, Pia Øllgaard Olsen, Nicole E. Blackburn, Dietrich Rothenbacher, Dhayana Dallmeier, Marta Roqué-Fíguls, Emma McIntosh, Carme Martín-Borràs

**Affiliations:** 1Department of Physical Activity and Sport Sciences, Faculty of Psychology, Education and Sport Sciences (FPCEE) Blanquerna, Ramon Llull University, Císter 34, 08022 Barcelona, Spain; mariagg@blanquerna.url.edu (M.G.-G.); myriamgb@blanquerna.url.edu (M.G.-B.); kellyprisciladf@blanquerna.url.edu (K.F.); mariacarmenmb@blanquerna.url.edu (C.M.-B.); 2Department of Physical Therapy, Faculty of Health Sciences (FCS) Blanquerna, Ramon Llull University, Padilla 326-332, 08025 Barcelona, Spain; 3Department of Physiatry and Nursing, Faculty of Health Sciences, University of Zaragoza, Domingo Miral, 50009 Zaragoza, Spain; brodriguez@posta.unizar.es; 4Sport and Exercise Sciences Research Institute, School of Sport, Ulster University, Newtownabbey BT37 0QB, UK; jj.wilson@ulster.ac.uk; 5Institute of Mental Health Sciences, School of Health Sciences, Ulster University, Newtownabbey BT37 0QB, UK; 6Center for Active and Healthy Ageing (CAHA), Department of Sports Science and Clinical Biomechanics, University of Southern Denmark, Campusvej 55, 5230 Odense, Denmark; PCaserotti@health.sdu.dk (P.C.); pollgaard@health.sdu.dk (P.Ø.O.); 7Institute of Nursing and Health Research, School of Health Sciences, Ulster University, Newtownabbey BT37 0QB, UK; ne.blackburn@ulster.ac.uk; 8Institute of Epidemiology and Medical Biometry, Ulm University, Helmholtztr 22, 89081 Ulm, Germany; Dietrich.Rothenbacher@uni-ulm.de; 9Research Unit on Aging, Agaplesion Bethesda Clinic, Zollernring 26, 89073 Ulm, Germany; dhayana.dallmeier@agaplesion.de; 10Department of Epidemiology, Boston University School of Public Health, 715 Albany Street, Boston, MA 02118, USA; 11Fundació Salut i Envelliment (Foundation on Health and Ageing)—UAB, Universitat Autònoma de Barcelona, Sant Antoni Maria Claret 171, 08041 Barcelona, Spain; Marta.Roque@uab.cat; 12Health Economics and Health Technology Assessment (HEHTA), Institute of Health and Wellbeing (IHW), University of Glasgow, Glasgow G12 8QQ, UK; Emma.McIntosh@glasgow.ac.uk

**Keywords:** sedentary behaviour, physical activity, accelerometer, health-related quality of life, older adults

## Abstract

Human movement behaviours such as physical activity (PA) and sedentary behaviour (SB) during waking time have a significant impact on health-related quality of life (HRQoL) in older adults. In this study, we aimed to analyse the association between self-reported and device-measured SB and PA with HRQoL in a cohort of community-dwelling older adults from four European countries. A subsample of 1193 participants from the SITLESS trial (61% women and 75.1 ± 6.2 years old) were included in the analysis. The association between self-reported and objective measures of SB and PA with HRQoL were quantified using Spearman’s Rho coefficients. The strength of the associations between self-reported and device-measured PA and SB with self-rated HRQoL (mental composite score, MCS; physical composite score, PCS) were assessed through multivariate multiple regression analysis. Self-reported and device-measured PA and SB levels showed significant but poor associations with PCS (*p* < 0.05). The association with MCS was only significant but poor with self-reported light PA (LPA) and moderate-to-vigorous PA (MVPA). In conclusion, the findings of this study suggest that both self-reported and device-measured PA of all intensities were positively and significantly associated, while SB was negatively and significantly associated with the PCS of the SF-12.

## 1. Introduction

Human movement behaviours such as physical activity (PA) and sedentary behaviour (SB) have a significant impact on health and quality of life (QoL) in older adults [1,2]. Health-related quality of life (HRQoL) in the older adult population is gaining the attention of researchers and policy makers since life expectancy worldwide, especially in European countries, is increasing. In 2015, the older adult population (≥65 years old) was 17.4% worldwide [3]; this percentage is expected to increase, reaching 33% by 2060 [4].

Low levels of SB, defined as any waking behaviour with an energy expenditure ≤1.5 Metabolic Equivalent Task (MET) while in a sitting, reclining or lying posture [5], and regular PA, have been associated with better-perceived quality of life in older adults [6,7,8,9,10]. More active and less sedentary older adults have shown better self-rated health in several studies [11,12]. Based on these findings, another recent study has suggested that higher levels of PA—and therefore, a better perception of HRQoL—were significantly associated with successful ageing measured with the Successful Ageing Scale for older adults [13]. Successful ageing is considered a complex construct cluster of factors such as QoL, life satisfaction and well-being, which includes movement behaviours [14]. Higher levels of PA predicted further maintenance of functional status including functional capacity, increased muscle mass and strength, which had also been related to HRQoL in this population [15]. Another study suggested that higher amounts of PA could improve cognitive and physical functions, leading to a positive effect on successful ageing in older adults [16]. High levels of SB had also been associated with HRQoL among older adults. Kim and colleagues showed that higher SB levels as older adults age were associated with poorer HRQoL [17]. A recent overview of systematic reviews showed that movement behaviours, including SB and PA levels, influenced heath status among adults [18]. SB and PA and their relationship to HRQoL have been widely studied in younger age groups (e.g., adolescents, adults) [19,20]. However, scientific literature on the older adult population is limited [21,22], and even more scarce using device-based measures of PA and SB in large samples [23].

Therefore, this study aimed to further investigate the associations between self-reported SB and PA and device-measured SB and PA with HRQoL in a cohort of community-dwelling older adults from four European countries.

## 2. Materials and Methods

### 2.1. Participants

This study used a cross-sectional design using baseline data from the SITLESS study. Briefly, the SITLESS study is a multi-centre, pragmatic, three-armed, randomised controlled trial. The purpose of the study was to determine whether an exercise referral scheme, enhanced by self-management strategies to reduce SB, could increase PA levels and improve health in the long term in community-dwelling European older adults (≥65 years old) from Spain, Germany, Denmark and United Kingdom (U.K.; in Northern Ireland). Included participants were insufficiently active (according to general guidelines) and/or reported being highly sedentary (>6 h in SB) [24]. The SITLESS study was approved by the Ethics and Research Committee of each institution. Participation was voluntary and all participants signed informed consent before the first assessment. Out of 1360 community-dwelling older adults, a subsample of 1193 participants who fully completed the SF-12, the SBQ, the modified IPAQ and returned valid accelerometer data from the hip-worn ActiGraph were analysed for this cross-sectional study.

### 2.2. Data Collection

Personal information including age, sex and educational background was collected with an individual interview and the number of current medications was obtained through the healthcare electronic registry. All demographic characteristics were collected during each assessment across all sites between 2016 and 2017. Weight and height were measured using a TANITA BC 420 and a SECA 213 portable stadiometer to obtain the participants’ body mass index (BMI). Participants were asked to self-report their daily sedentary time using the SBQ [25] and their total time spent walking (LPA; 3.3 METs per minute), in moderate PA (MPA; 4 METs per minute) and in vigorous PA (VPA; 8 METs per minute) using the modified IPAQ [26]. Both questionnaires are valid to assess SB [27] and PA [28]. Some limitations such as recall bias and poor correlation with objective measures are likely with self-reported measures of both behaviours in older adults [27,29,30,31,32]. However, they have been widely used in large-scale studies due to their low administration cost. Moreover, the SBQ provides relevant information regarding context of behaviour [33]. Self-rated HRQoL was assessed using the 12-Item Short-Form Health Survey (SF-12) to obtain the physical and mental composite scores [34]. The SF-12 questionnaire has been used as an important tool to describe self-reported perception of HRQoL [35]. The SF-12 scoring method proposed by Ware et al. (1996) assumes that each item (from an 8-dimension profile including physical functioning, role limitations due to physical problems, bodily pain, general health, vitality, limitations due to emotional problems and mental health) contributes to both the physical component score (PCS) or the mental component score (MCS), and that these two measures are uncorrelated [34]. To overcome the aforementioned limitations of self-reported measures, participants were also asked to wear an ActiGraph wGT3X-BT triaxial accelerometer (ActiGraph, LLC, Pensacola, FL, USA) on their dominant hip during waking hours for seven consecutive days. Participants were asked to remove the device only for water-based activities (e.g., bathing or swimming) and during sleep time. Wear time and non-wear time was calculated using the Choi 2011 algorithm [36]. A small number of participants wore the device continuously (i.e., no removal during sleep). To reduce conflation of sleep and SB time, a pragmatic maximum daily wear time threshold of 19 h was used. For relevant participants, their activity diary was used to record on/off times. The accelerometers were initialised to collect data at 30 Hz using the normal filter setting. A valid accelerometer dataset contained at least four valid days (including at least one weekend day), with a valid day defined as containing at least 600 min of wear time to be included in the analysis as in previous studies [37]. SB was defined as <100 counts per minute (CPM), LPA as 100–2019 CPM and MVPA as ≥2020 CPM on the vertical axis [38]. Raw accelerometry data were analysed using ActiLife v6.13.3 software summarised into 10 s epochs, as has been recommended for estimation of SB in clinical older adult populations [39].

### 2.3. Statistical Analysis

Demographic characteristics of the sample size were presented descriptively as mean and standard deviation (SD) for continuous variables and percentage for categorical variables, separately by country and overall. The association between self-reported and objective measures of SB and PA and HRQoL were quantified using non-parametric Spearman’s Rho coefficients after all variables were examined for normality using the Kolmogorov–Smirnov test. Correlations were interpreted as follows: coefficient value between +1 and −1, perfect positive/negative linear relationship or correlation; between +0.8 and −0.9, very strong positive/negative linear relationship or correlation; between +0.6 and −0.7, moderate positive/negative linear relationship or correlation; between +0.3 and −0.5, fair positive/negative linear relationship or correlation; between +0.1 and −0.2, poor positive/negative linear relationship or correlation; 0, non-linear relationship or correlation [40]. The strength of the association between self-reported time spent in PA and SB (model *a*) and device-measured daily time spent in PA and SB (model *b*) and self-rated HRQoL (SF-12: MCS and PCS) were assessed through multivariate multiple regression analysis. Model *c* and model *d* were adjusted for covariates: country (Spain, Germany, Denmark and U.K.), age (years), sex (male/female), BMI categories (≤24.9 underweight and normal, 25.0–29.9 overweight, ≥30 obese), educational background (never attended school, primary education, secondary education, university, unknown) and number of current medications. Each independent variable was investigated for collinearity using the variance inflation factor (VIF; collinearity: VIF > 4), which identifies correlation between independent variables and the strength of that correlation. All statistical analyses were performed using IBM SPSS Statistics 26 (SPSS, Inc., an IBM Company, Chicago, IL, USA) and the significance level was set at *p* < 0.05.

## 3. Results

Out of the 1360 participants in the SITLESS trial, 167 participants were excluded as they did not meet the pre-specified ActiGraph wear time criteria or did not complete the self-reported questionnaires. A final subsample of 1193 participants (75.1 ± 6.2 years old, 61% women) returned valid accelerometer data, completed the SF-12, the SBQ and the IPAQ.

Descriptive characteristics of the sample are presented in Table 1. Approximately 75% of the overall sample was overweight or obese (mean BMI = 28.8 ± 5.2 and ranged from 16.7 kg/m^2^ to 51.5 kg/m^2^). Women BMI ranged from 16.7 kg/m^2^ to 51.5 kg/m^2^ (mean women BMI = 28.9 ± 5.8) and men BMI ranged from 19.8 kg/m^2^ to 45.8 kg/m^2^ (mean men BMI = 28.7 ± 4.2). Of all participants, 54.1% reported having completed secondary education while 3.1% of the overall sample reported that they never attended school. The number of current medications ranged from 0 to 19 with a mean of 4.5 ± 3.2. Some 70% of the participants reported good to excellent general health. Nevertheless, a small percentage (3.9%) reported poor general health status with the SF-12 survey. The PCS across all participants (45.0 ± 9.1) ranged from 15.6 to 65.7. The MCS across all participants (51.9 ± 8.9) ranged from 18.4 to 71.1.

Descriptive characteristics of self-reported and device-measured SB and PA levels are shown in Table 1. No significant differences (*p* = 0.082) were found between countries regarding the self-reported average mean hours per day of SB (overall mean average 7.7 ± 2.8 h/d). Self-reported LPA differed (*p* < 0.05 using Bonferroni’s test) between Spanish (1.0 ± 1.3 h/d) and U.K. participants (1.0 ± 1.0 h/d) versus Danish (1.3 ± 1.6 h/d) and German participants (1.3 ± 1.4 h/d). U.K. participants showed significant differences (*p* < 0.05 using Bonferroni’s test) against other countries’ participants regarding MVPA levels (overall 0.7 ± 1.1 h/d). Device-measured SB was 78.8% of daily awake time out of 14.4 h of mean daily wear time in the overall sample. Danish participants showed the highest proportion of device-measured daily SB (81.0%). Device-measured daily LPA was 18.6% and MVPA was 2.6% in the overall sample. U.K. participants showed the highest levels on both PA intensities (19.8% and 3.7%, respectively).

Table 2 displays the association between self-reported and device-measured levels of PA and SB with PCS as well as MCS. PA and SB levels, both device-measured and self-reported, showed a poor-to-fair significant association (*p* ≤ 0.05 across all PA and SB variables) with PCS. It is important to note that all SB-related measures (both device-based assessment and self-reported) showed a poor negative significant association with PCS in the model (e.g., the less sedentary an individual was, the better PCS was perceived) (*p* < 0.05). Nevertheless, when the association between self-reported and device-measured levels of PA and SB with MCS was analysed, significant associations were only found between self-reported LPA and MVPA (*p* < 0.05), but no significant associations were found between device-measured daily hours in LPA and MVPA or daily hours in SB with MCS.

Table 3 shows the multivariate multiple regression models for PCS and MCS, unadjusted (models a and b) and adjusted for relevant covariates (models c and d). The full model a for the PCS adjusted by self-reported PA and SB time predicted 19% of the total variance (*p* < 0.001). The effect modification was significant (*p* < 0.05) in all explanatory self-reported variables except for LPA time (IPAQ). The same model for the MCS predicted 0.07% of the total variance (*p* < 0.001) with a significant effect modification in self-reported daily sedentary time (SBQ) (*p* < 0.05). No collinearity was identified between each independent variable of this model (VIF ranges from 1.02 to 1.15). The full model b for PCS adjusted by device-measured SB and PA (LPA and MVPA) predicted 14% of the total variance (*p* < 0.001) with a significant effect modification (*p* < 0.05) in device-measured LPA, MVPA and SB. On the other hand, the full model for MCS adjusted for device-measured SB and PA (LPA and MVPA) predicted 0.1% of the total variance (*p* < 0.001) with a significant effect modification (*p* < 0.05) in MVPA and sedentary time. No collinearity was identified between each independent variable of this model (VIF ranges from 1.10 to 1.47). The model c for PCS and MCS adjusted by self-reported PA and SB, and for covariates predicted 25% and 0.8% of the total variance (*p* < 0.001), respectively. The model d for the same dependent variables was adjusted by device-measured SB and PA, and for covariates predicted 29% of the total variance for PCS and 0.08% for MCS (*p* < 0.001).

## 4. Discussion

This study aimed to analyse the association between self-reported and device-measured SB and PA with HRQoL in a cohort of community-dwelling older adults from four European countries, assessing possible differences between the PCS and the MCS of the SF-12 questionnaire. Our results showed poor-to-fair significant associations between self-reported and device-measured SB and PA with the PCS. When assessing these associations with MCS, poor significant associations were found with self-reported LPA and MVPA. Our multivariate multiple regression models adjusted by self-reported and device-measured PA and SB predicted between 19% and 14% of the variance in the PCS by both self-reported and device-measured SB and PA. For MCS, the same models, self-reported and device-measured PA and SB, predicted between 0.07% and 0.1% of the variance.

A recent study that analysed self-reported PA levels with HRQoL using the SF-12 found that more active participants reported higher levels of HRQoL as well as significantly higher MCS (*p* < 0.01) [8]. This significant association of self-reported PA with MCS differs from our findings, and could be potentially explained due to a difference in the age of the sample and the use of a different questionnaire which assessed several types of PA and different intensities using METs [8]. Another recent study found that self-reported physical inactivity was significantly associated with diminished HRQoL, including physical and mental health composite scores [41]. In our study, device-measured PA levels were only significantly associated with PCS. A recent study that aimed to assess the association of device-measured PA levels of different intensities (light, moderate and MVPA) in women with fibromyalgia found that all PA intensity levels were positively associated with HRQoL [42]. Our study found a higher association with device-measured MVPA and PCS than in the al-Ándalus project [42]. This higher association found in our study could be explained due to a worse baseline functional state in the al-Ándalus project’s population (mean PCS 29.5 ± 6.9 versus PCS 45.0 ± 9.1 units), which may prevent their participants from engaging in higher-intensity PA. Davis and colleagues (2014) found that higher levels of device-based MVPA and greater SB were associated with greater and worse Short Physical Performance Battery scores, respectively, in adults >70 years old [43]. Previous research has suggested that engaging in MVPA could contribute to better physical function performance in older adults, which is related to some of the eight dimensions assessed in the SF-12 to assess PCS, such as physical functioning or limitations due to physical problems [44,45,46]. However, taking into account the cross-sectional nature of these data, such associations could reflect bi-directional causation, where a better physical function is needed to engage in most types of MVPA.

Previous scientific literature suggests that PA and SB are independently associated with HRQoL, and that results regarding SB are still mostly inconsistent [47]. In our study, we found a significant, negative and poor association between self-reported SB and PCS. López-Torres and colleagues (2019) also concluded that higher self-reported SB levels were significantly associated with poorer HRQoL. Likewise, Wilson and colleagues [10] found that daily sitting time was negatively and significantly associated with HRQoL. Similarly, other recent studies found significant associations between device-measured SB and HRQoL [42,48,49,50,51]. However, previous studies suggested that device-measured PA levels tend to be more valid and reliable than self-reported [38,52]. Therefore, the lower associations between self-reported SB and PA levels with PCS compared to device-measured SB and PA levels could be explained by the accuracy in measuring both behaviours. Nevertheless, having analysed the associations between SB and PA between MCS, only significant but poor associations were found with self-reported PA levels.

The model used to predict MCS from the al-Ándalus project had higher predictive capability (R^2^ = 0.19) compared to our results in both SB and PA models with the MCS [42]. However, the PCS models with both self-reported and device-measured SB and PA showed higher prediction of the total variance compared to the al-Ándalus project (R^2^ = 0.04). As previously stated, the different regression coefficients between studies could be explained by the different sample characteristics (e.g., the sample with women suffering from fibromyalgia included in the al-Ándalus project reported lower PCS than older adults from the SITLESS study) or the instrument used to assess HRQoL. Our results may not be clinically significant due to some of the correlations being poor. However, these findings may provide insight when designing strategies and interventions aimed at improving both physical and mental components of overall quality of life. Furthermore, we must suggest that public resources should be allocated to strategies aimed at reducing SB and increasing PA levels to improve quality of life of older adults.

### Strengths and Limitations

Some limitations of the present cross-sectional study are worth noting. It is difficult to determine associations between physical behaviours and HRQoL due to the large number of factors related to HRQoL [53]. The strengths of this study include the device-measured SB and PA of a large heterogeneous sample of community-dwelling older adults from four European countries as well as their association with HRQoL in both the physical and mental components.

## 5. Conclusions

In conclusion, the findings of this study suggest that both self-reported or device-measured PA intensity levels were positively and statistically significant associated with PCS of the SF-12. On the other hand, SB was negatively and statistically significant associated with the PCS of the SF-12. These findings indicate the importance of increasing PA and reducing SB levels if the physical function component related to HRQoL is to be improved. Only self-reported LPA and MVPA were statistically significant with MCS. In general, some of the associations we found were very poor, but our results support the accumulating research of the benefits of increasing PA levels and reducing sitting time for better HRQoL among older adults. Further longitudinal studies are necessary to confirm the associations between the PCS and self-rated health and further explore associations between the MCS.

## Figures and Tables

**Table 1 ijerph-18-13252-t001:** Descriptive variables of the sample.

	Overall (*n* = 1193)	Spain (*n* = 263)	Germany (*n* = 304)	Denmark (*n* = 318)	U.K. (*n* = 308)
Age (years), mean (SD)	75.2 (6.2)	75.9 (6.4)	74.7 (6.1)	77.3 (5.7)	72.6 (5.5)
Sex (women), *n* (%)	733 (61.4)	207 (78.7)	172 (56.6)	185 (58.2)	169 (54.9)
BMI, mean (SD)	28.8 (5.2)	28.9 (5.0)	29.2 (5.6)	27.3 (5.0)	28.9 (5.0)
BMI categories, *n* (%)					
Underweight and normal (≤24.9 kg/m^2^)	274 (23.0)	39 (14.8)	61 (20.1)	106 (33.4)	68 (22.1)
Overweight (25 to 29.9 kg/m^2^)	488 (40.9)	102 (38.8)	131 (43.1)	129 (40.7)	126 (40.9)
Obese (≥30 kg/m^2^)	430 (36.1)	122 (46.4	112 (36.8)	82 (25.9)	114 (37.0)
Educational background, *n* (%)					
Never attended school	37 (3.1)	35 (13.4)	1 (0.3)	1 (0.3)	-
Primary education	232 (19.5)	112 (42.9)	9 (3.0)	93 (29.3)	18 (5.9)
Secondary education	642 (54.1)	84 (32.2)	217 (71.9)	182 (57.4)	159 (51.8)
University	272 (22.9)	29 (11.1)	74 (24.5)	40 (12.6)	129 (42.0)
Unknown	4 (0.4)	3 (0.4)	3 (0.3)	2 (0.3)	2 (0.3)
Number of current medications, mean (range)	4.5 (0–19)	4.8 (0–16)	4.3 (0–16)	3.9 (0–14)	4.8 (0–19)
Self-reported general health (SF-12), *n* (%)					
Excellent	35 (2.9)	1 (0.4)	1 (0.3)	16 (5.0)	17 (5.5)
Very good	216 (18.1)	19 (7.2)	24 (7.9)	69 (21.7)	104 (33.8)
Good	584 (49.0)	116 (44.1)	160 (52.6)	177 (55.7)	131 (42.5
Fair	311 (26.1)	106 (40.3)	107 (35.2)	51 (16.0)	47 (15.3)
Poor	47 (3.9)	21 (8.0)	12 (3.9)	5 (1.6)	9 (2.9)
Physical component (SF-12), mean (SD)	45.0 (9.1)	42.1 (9.4)	43.6 (8.7)	46.1 (8.6)	47.6 (8.9)
Mental component (SF-12), mean (SD)	51.9 (8.9)	49.7 (9.1)	50.1 (9.3)	54.1 (8.1)	53.2 (8.2)
Self-reported SB (h/d), mean (SD)					
Daily sedentary time	7.7 (2.8)	7.4 (3.2)	7.6 (2.4)	7.9 (2.7)	7.9 (2.8)
IPAQ (h/d), mean (SD)					
LPA	1.9 (1.3)	1.0 (1.0)	1.3 (1.4)	1.3 (1.6)	1.0 (1.0)
MVPA	0.6 (1.1)	1.1 (1.1)	0.6 (0.9)	0.5 (0.8)	1.1 (1.2)
Accelerometry (h/d), mean (SD)					
Daily sedentary time	11.3 (1.3)	11.3 (1.3)	11.2 (1.3)	11.7 (1.1)	10.9 (1.1)
Daily LPA	2.7 (0.9)	2.6 (0.8)	2.7 (0.9)	2.5 (0.9)	2.8 (0.8)
Daily MVPA	0.4 (0.3)	0.3 (0.3)	0.4 (0.3)	0.2 (0.2)	0.5 (0.4)
Total wear time (h/d), mean (SD)	14.4 (1.2)	14.2 (1.2)	14.3 (1.2)	14.5 (1.1)	14.3 (1.0)

BMI: body mass index; IPAQ: international physical activity questionnaire; LPA: light physical activity; MVPA: moderate-vigorous physical activity.

**Table 2 ijerph-18-13252-t002:** Relationship between self-reported and device-measured SB and PA, and the physical and mental component scores of the SF-12 (HRQoL). Spearman’s Rho coefficients and *p*-values.

	Physical Component SF-12		Mental Component SF-12	
	Spearman’s Rho	*p*-Value	Spearman’s Rho	*p*-Value
Self-reported SB (SBQ)				
Daily sedentary time	−0.078 **	0.007	0.033	0.251
Self-reported PA (IPAQ)				
LPA	0.091 **	0.002	0.058 *	0.046
MVPA	0.204 **	<0.001	0.072 *	0.013
Accelerometry				
Daily Sedentary Time	−0.263 **	<0.001	0.012	0.674
Daily LPA	0.168 **	<0.001	−0.029	0.319
Daily MVPA	0.419 **	<0.001	0.051	0.081

SBQ: sedentary behaviour questionnaire; IPAQ: international physical activity questionnaire; MVPA: moderate-vigorous physical activity; LPA: light physical activity. ** *p* ≤ 0.001; * *p* ≤ 0.05.

**Table 3 ijerph-18-13252-t003:** Multivariate multiple regression models for PCS and MCS adjusted by self-reported and device-measured PA and SB levels.

	PCS			MCS		
	R^2^	Non St. Beta	*p*-Value	R^2^	Non St. Beta	*p*-Value
Model *a*	0.19		<0.001	0.007		<0.001
LPA (IPAQ)		0.198	0.322		0.316	0.105
MVPA (IPAQ)		0.735	0.002		0.071	0.763
Daily sedentary time (SBQ)		−0.313	0.001		−0.200	0.028
Model *b*	0.148		<0.001	0.016		<0.001
LPA (ActiGraph)		0.793	0.019		0.161	0.647
MVPA (ActiGraph)		10.210	<0.001		2.564	0.003
Sedentary time (ActiGraph)		0.637	0.005		0.880	<0.001
Model *c*	0.258	-	<0.001	0.081	-	<0.001
LPA (IPAQ)		−0.047	0.797		0.297	0.133
MVPA (IPAQ)		0.412	0.063		−0.078	0.748
Daily sedentary time (SBQ)		−0.244	0.004		−0.220	0.017
Country (Denmark as reference)						
Spain		−2.284	0.002		−3.464	<0.001
UK		1.163	0.103		−0.919	0.237
Germany		−2.705	<0.001		−4.334	<0.001
Age (years)		−0.197	<0.001		0.070	0.120
Sex (women as reference)						
Men		1.589	0.002		2.101	<0.001
BMI categories (obese as reference)						
Underweight and normal		2.666	<0.001		−1.515	0.038
Overweight		3.226	<0.001		−0.070	0.908
Educational background (university as reference)						
Never attended school and primary		−0.650	0.418		−1.778	0.042
Secondary		0.045	0.766		−0.225	0.173
Number of current medications		−0.976	<0.001		−0.213	0.010
Model *d*	0.293	-	<0.001	0.082	-	<0.001
LPA (ActiGraph)		0.389	0.244		0.493	0.187
MVPA (ActiGraph)		6.434	<0.001		1.825	0.011
Sedentary time (ActiGraph)		0.284	0.189		0.109	0.059
Country (Denmark as reference)						
Spain		−2.728	<0.001		−3.373	<0.001
UK		0.152	0.831		−1.056	0.184
Germany		−3.183	<0.001		−4.178	<0.001
Age (years)		−0.088	0.040		0.101	0.034
Sex (women as reference)						
Men		1.368	0.007		2.123	<0.001
BMI categories (obese as reference)						
Underweight and normal		1.623	0.015		−1.663	0.026
Overweight		2.511	<0.001		−0.245	0.058
Educational background (university as reference)						
Never attended school and primary		−0.492	0.531		−1.667	0.058
Secondary		0.086	0.559		−0.218	0.186
Number of current medications		−0.832	<0.001		−0.180	0.035

Abbreviations: PCS, physical composite score; MCS, mental composite score; R^2^, r square; Non St Beta., non-standardised beta coefficient; SBQ, sedentary behaviour questionnaire; IPAQ, international physical activity questionnaire; LPA, light physical activity; MVPA, moderate to vigorous physical activity. *a* PCS and MCS adjusted by self-reported PA (LPA and MVPA) and SB. *b* PCS and MCS adjusted by device-measured PA (LPA and MVPA) and SB. *c* PCS and MCS adjusted by self-reported PA (LPA and MVPA), SB and covariates. *d* PCS and MCS adjusted by device-measured PA (LPA and MVPA), SB and covariates.

## Data Availability

Not applicable.

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
