# Peer review of "Association of Self-Reported and Device-Measured Sedentary Behaviour and Physical Activity with Health-Related Quality of Life among European Older Adults"

_ijerph, 2021, doi:10.3390/ijerph182413252_

Round 1

Reviewer 1 Report

This cross-sectional study investigates the associations between self-reported and device-based sedentary behavior and physical activity with health-related quality of life among older adults. The study benefits from a relatively large sample of community-dwelling older adults from four European countries and the use of both self-reported and device-based measures of daily sedentary and physical activity time. The research question is relevant, yet the rationale and reporting of the study need some critical corrections.

Introduction:

  • The authors should give a better justified rationale, why it is important to investigate the associations of physical activity and sedentary behavior with self-rated health-related quality of life. The introduction should concentrate more on how and why physical activity and sedentary behavior may be important for physical and mental health related quality rather than on describing the methodological issues to assess PA, SB and HRQoL

Methods:

  • Please describe the utilized questionnaires rather here than in the introduction.
  • Justify, why percentages chosen for accelerometer-based SB and PA and not minutes/hours per day? Min/h would be better comparable to self-reports.
  • Consider analyzing consistency of self-reported and accelerometer-based SB and PA. How strong are the associations between self-reported and accelerometer-measured SB/LPA/MVPA? These results would be interesting, and allow discussing the different results for self-reported and accelerometer-based SB and PA. The introduction and reference list concentrate much on the use of self-report vs. device-based measures and their validity, but the analyses and discussion do not contribute to this point of view.
  • Please justify why Spearmans Rho was chosen and not Pearson’s correlation coefficient?
  • Justify the selection of covariates in the regression models and consider presenting unadjusted models (including only SB & PA variables).
  • Consider performing sensitivity analyses and presenting results for each country separately due to differences in PA levels and significance of setting in the regression models. Country-specific information could be provided as supplementary data.

Results:

  • Many of the covariates are categorical, but it seems that they have been treated as continuous variables in the regression analyses. Table 3 is confusing as for the covariates, how should the beta coefficients be interpreted for e.g., setting? Furthermore, consider if it is necessary to show the values for covariates – they are not in the focus of this study.
  • Additionally, consider presenting also results from unadjusted models, i.e., regression models including only the SB and PA variables.
  • What was the proportion of variance explained by SB and PA variables is more interesting and answers the research question better than the whole model including covariates. Some of the bivariate correlations are statistically significant, but very weak (<0.1), thus the covariates may explain a much bigger proportion of the variance than the SB and PA variables, that should be the main interest.

Discussion:

  • The discussion is lengthy and concentrates too much on other studies’ findings. The authors should focus more on their own findings than reporting results from previous studies. The discussion should focus on the main findings and their relevance, not repeat the results. Any p-values or other exact results should not be reported in the discussion.
  • Using both self-reported and accelerometer-based measures of PA and SB is one of the strengths of the study. It would be important to discuss, why the findings were different for self-reported and accelerometer-based PA and SB.
  • It is not clear in the discussion when the authors refer to the unadjusted correlations and when to the adjusted regression models. What are the main findings? This should be clarified. Furthermore, some of the correlations were very weak (p<0.1) and thus probably not clinically significant even though statistically significant. Please consider this in the discussion.

Conclusions:

  • Some of the associations were very weak and this was a cross-sectional study. The conclusions about the importance of increasing PA and reducing SB for improving physical HRQoL are thus too strong. “Significantly associated” is not a correct phrasing here – the associations were statistically significant, yet the clinical significance is questionable.
  •  

Minor

  • L62ff: Define also the different intensities of PA with the help of METs. These definitions would fit better in the methods section.
  • L71–72: How are the 24h activity cycle and sleep relevant to the present study?
  • L81: Self-report questionnaires to assess what?
  • L 81-82, 87-88: It is not necessary to name the questionnaires and accelerometer brands in the introduction.
  • L88: For a more balanced introduction, describe the imitations regarding the use of accelerometers, especially among older adults.
  • L100-106: Please rephrase, this sentence is over long and difficult to read.
  • L112-113: Were the background characteristics self-reported or register-based?
  • L139-142: The interpretation of the correlations is very imprecise in my opinion. ±5 to ±1 could be anything between -0.5 to 1 or 0.5 to -1.
  • L146-147: Justify, why BMI was categorized and not used as a continuous variable? And, if categorized, why were underweight and normal weight participants combined into one category, while it is known that underweight is a health risk for older adults?
  • L160: Table 1 includes also other than demographic characteristics, please rephrase.
  • Table 2: SB during weekdays and weekend days is not described in the methods nor included in the regression models or discussed. Could be removed, does not add value.
  • L252-254: The proportion of variance explained by the regression models is not interesting, while it does not respond to the research question. Please concentrate on the role of PA and SB.
  • L281ff it is unclear, how the referenced studies investigating the associations between physical function/performance tests and PA relate to the present study with self-reported HRQoL.
  • L337: Associated with what?

Author Response

Comments and Suggestions for Authors

This cross-sectional study investigates the associations between self-reported and device-based sedentary behavior and physical activity with health-related quality of life among older adults. The study benefits from a relatively large sample of community-dwelling older adults from four European countries and the use of both self-reported and device-based measures of daily sedentary and physical activity time. The research question is relevant, yet the rationale and reporting of the study need some critical corrections.

Introduction:

The authors should give a better justified rationale, why it is important to investigate the associations of physical activity and sedentary behavior with self-rated health-related quality of life. The introduction should concentrate more on how and why physical activity and sedentary behavior may be important for physical and mental health related quality rather than on describing the methodological issues to assess PA, SB and HRQoL.

The authors thank the reviewer for his/her feedback. We have added more rationale that highlights the importance of physical activity and sedentary behaviour in improving HRQoL in the introduction section, and have deleted some information more focused in methodological issues to assess different behaviours and health status.

Methods:

Please describe the utilized questionnaires rather here than in the introduction.

The authors appreciate your comment and had moved the explanation of the questionnaires in the methods section.

Justify, why percentages chosen for accelerometer-based SB and PA and not minutes/hours per day? Min/h would be better comparable to self-reports.

The authors appreciate your comment and have changed percentages of accelerometer-based SB and PA into hours per day.

Consider analyzing consistency of self-reported and accelerometer-based SB and PA. How strong are the associations between self-reported and accelerometer-measured SB/LPA/MVPA? These results would be interesting, and allow discussing the different results for self-reported and accelerometer-based SB and PA. The introduction and reference list concentrate much on the use of self-report vs. device-based measures and their validity, but the analyses and discussion do not contribute to this point of view.

The authors appreciate your comment. We agree that consistency of self-reported and accelerometer-based SB and PA would be interesting. However, it was not the aim of the present study. Consistency of self-reported and accelerometer-based SB and their associations have been analyzed using the data in another study when we validated the SBQ in three languages against the accelerometer data as a gold standard measure.

Sansano-Nadal, O.; Wilson, J.J.; Martín-Borràs, C.; Brønd, J.C.; Skjødt, M.; Caserotti, P.; Roqué i Figuls, M.; Blackburn, N.E.; Klenk, J.; Rothenbacher, D.; et al. Validity of the Sedentary Behavior Questionnaire in European Older Adults Using English, Spanish, German and Danish Versions. Meas. Phys. Educ. Exerc. Sci. 2021, 1–14, doi:10.1080/1091367X.2021.1922910.

Please justify why Spearmans Rho was chosen and not Pearson’s correlation coefficient?

The authors appreciate your comment and have justified why we used Spearmans Rho in the manuscript. Before conducting analyses, all variables were examined for normality using kolmogorov-Smirnov Test. After the normality analysis, we used non-parametric tests.

Justify the selection of covariates in the regression models and consider presenting unadjusted models (including only SB & PA variables).

The authors appreciate your comment and had added two new unadjusted regression models including only SB and PA variables in Table 3.

Consider performing sensitivity analyses and presenting results for each country separately due to differences in PA levels and significance of setting in the regression models. Country-specific information could be provided as supplementary data.

The authors thank you for your comment. Results for each country separately due to the differences in PA and SB levels using self-reported and accelerometer-based data have been published elsewhere using the data (see the references). However, in Table 1 we had included descriptive variables for each country separately.

Giné-Garriga, M.; Sansano-Nadal, O.; Tully, M. A.; Caserotti, P.; Coll-Planas, L.; Rothenbacher, D.; Dallmeier, D.; Denkinger, M.; Wilson, J. J.; Martin-Borràs, C.; Skjødt, M.; Ferri, K.; Farche, A. C.; McIntosh, E.; Blackburn, N. E.; Salvà, A.; Roqué-i-Figuls, M. Accelerometer-Measured Sedentary and Physical Activity Time and Their Correlates in European Older Adults: The SITLESS Study. Journals Gerontol. Ser. A 2020. https://doi.org/10.1093/gerona/glaa016.

Sansano-Nadal, O.; Wilson, J.J.; Martín-Borràs, C.; Brønd, J.C.; Skjødt, M.; Caserotti, P.; Roqué I Figuls, M.; Blackburn, N.E.; Klenk, J.; Rothenbacher, D.; et al. Validity of the Sedentary Behavior Questionnaire in European Older Adults Using English, Spanish, German and Danish Versions. Meas. Phys. Educ. Exerc. Sci. 2021, 1–14, doi:10.1080/1091367X.2021.1922910.

Results:

Many of the covariates are categorical, but it seems that they have been treated as continuous variables in the regression analyses. Table 3 is confusing as for the covariates, how should the beta coefficients be interpreted for e.g., setting? Furthermore, consider if it is necessary to show the values for covariates – they are not in the focus of this study.

The authors appreciate your comment and had deleted values for covariates in Table 3, accordingly.

Additionally, consider presenting also results from unadjusted models, i.e., regression models including only the SB and PA variables.

What was the proportion of variance explained by SB and PA variables is more interesting and answers the research question better than the whole model including covariates. Some of the bivariate correlations are statistically significant, but very weak (<0.1), thus the covariates may explain a much bigger proportion of the variance than the SB and PA variables, that should be the main interest.

The authors thank you for your suggestion and had added unadjusted models, including only SB and PA variables in Table 3 (see model a and b). We have also maintained the two models adjusted by a selection of covariates, which had been shown to affect SB and PA in previous studies.

Discussion:

The discussion is lengthy and concentrates too much on other studies’ findings. The authors should focus more on their own findings than reporting results from previous studies. The discussion should focus on the main findings and their relevance, not repeat the results. Any p-values or other exact results should not be reported in the discussion.

The authors appreciate your comments. We had tried to reduce the length of the discussion section. We have also tried to focus more in our findings without being reiterative of the results section.

Using both self-reported and accelerometer-based measures of PA and SB is one of the strengths of the study. It would be important to discuss, why the findings were different for self-reported and accelerometer-based PA and SB.

The authors appreciate your comments and had tried to discuss the findings regarding the differences on the associations between PCS / MCS and self-reported and device-measured SB / PA. 

It is not clear in the discussion when the authors refer to the unadjusted correlations and when to the adjusted regression models. What are the main findings? This should be clarified. Furthermore, some of the correlations were very weak (p<0.1) and thus probably not clinically significant even though statistically significant. Please consider this in the discussion.

The authors appreciate your comments and had corrected the section when compared the unadjusted regression models with the Al-Andalus project. 

Conclusions:

Some of the associations were very weak and this was a cross-sectional study. The conclusions about the importance of increasing PA and reducing SB for improving physical HRQoL are thus too strong. “Significantly associated” is not a correct phrasing here – the associations were statistically significant, yet the clinical significance is questionable.

The authors appreciate your comments and had modified the conclusions section, accordingly.

Minor

L62ff: Define also the different intensities of PA with the help of METs. These definitions would fit better in the methods section.

The authors appreciate your comment and have defined different intensities of PA assessed with the modified IPAQ with the help of METs in the methods section.

L71–72: How are the 24h activity cycle and sleep relevant to the present study?

The authors appreciate your comment and have deleted this information following your suggestion that the 24 hours activity cycle is not relevant to the present study.

L81: Self-report questionnaires to assess what?

The authors appreciate your comment and had corrected the text, accordingly.

L 81-82, 87-88: It is not necessary to name the questionnaires and accelerometer brands in the introduction.

The authors appreciate your comment. Name of the questionnaires and accelerometer brand are not in the introduction section as we have modified it accordingly to your suggestions.

L88: For a more balanced introduction, describe the limitations regarding the use of accelerometers, especially among older adults.

Following your previous suggestions, we have moved the limitations of the questionnaires in the methods section.

L100-106: Please rephrase, this sentence is over long and difficult to read.

The authors appreciate your comment and had corrected the text, accordingly (Line 89).

L112-113: Were the background characteristics self-reported or register-based?

The authors thank you for your comment and had added this information.

L139-142: The interpretation of the correlations is very imprecise in my opinion. ±5 to ±1 could be anything between -0.5 to 1 or 0.5 to -1.

The authors appreciate your comment and had changed the interpretation of the correlations following:  coefficient value between +1 to -1: perfect positive/negative linear relationship or correlation; between +0.8 to +0.9: very strong positive/negative linear relationship or correlation; between +0.6 to -0.7: moderate positive/negative linear relationship or correlation; between +0.3 to -0.5 fair positive/negative linear relationship or correlation; between +0.1 to -0.2: poor positive/negative linear relationship or correlation; and 0: non linear relationship or correlation.

Reference: Akoglu, H. User’s guide to correlation coefficients. Turkish J. Emerg. Med. 2018, 18, 91, doi:10.1016/J.TJEM.2018.08.001.

L146-147: Justify, why BMI was categorized and not used as a continuous variable? And, if categorized, why were underweight and normal weight participants combined into one category, while it is known that underweight is a health risk for older adults?

The authors appreciate your comment. We had shown BMI as a continuous variable as well as categorized (Table 1). Underweight and normal weight participants were combined as we found that just 4 participants (0.3% of the sample) were underweight.

L160: Table 1 includes also other than demographic characteristics, please rephrase.

The authors appreciate your comment and had corrected the text, accordingly.

Table 2: SB during weekdays and weekend days is not described in the methods nor included in the regression models or discussed. Could be removed, does not add value.

The authors appreciate your comment and had deleted SB during weekdays and weekend days as suggested.

L252-254: The proportion of variance explained by the regression models is not interesting, while it does not respond to the research question. Please concentrate on the role of PA and SB.

The authors appreciate your comment and had added just the proportion of variance explained by PA and SB.

L281ff it is unclear, how the referenced studies investigating the associations between physical function/performance tests and PA relate to the present study with self-reported HRQoL.

The authors appreciate your comment and had justified the relationship of the associations between physical function/performance tests and PA with HRQoL (in line 297-299). Physical function and limitations due to physical problems are two of the 8 dimensions to assess PCS using the SF-12.

L337: Associated with what?

The authors appreciate your comment and had corrected the text, accordingly.

Reviewer 2 Report

Thank you for an interesting article which I have enjoyed reading. Please see suggestions for clarification/improvement.

Line 70- the idea of 24 hr activity cycle have been introduced for a while now- at least 2014, See Chaput et al., (2014) https://pubmed.ncbi.nlm.nih.gov/25485978/

Line 112- dates/years for data collection missing. Please add.

Line 123- check spelling + the fact that some wore the device continuously might be better suited in results.

Line 125- what hours were labelled as sleep and was there any evidence used to inform this?

Line 132- were there any measures used to identify non-wear time?

Line 138- could the authors provide some information about PA data distribution.

Line 155- are the authors reporting on secondary data analyses- this is unclear

Line 162- could the authors provide obesity results by sex

Line 182- a percentage breakdown of participants by country would be informative. Perhaps another table?

Line 325/326 this is an expected finding and previously reported. Do the authors believe we still need to further investigate this cross -sectionally or what do they advise?

Author Response

Comments and Suggestions for Authors

Thank you for an interesting article which I have enjoyed reading. Please see suggestions for clarification/improvement.

The authors thank the reviewer for his/her positive feedback. We have included his/her suggestions in the manuscript.

Line 70- the idea of 24 hr activity cycle have been introduced for a while now- at least 2014, See Chaput et al., (2014) https://pubmed.ncbi.nlm.nih.gov/25485978/

The authors have deleted the information regarding the 24 hours activity cycle as the reviewer 1 had suggested that this information is not relevant to the present study.

Line 112- dates/years for data collection missing. Please add.

The authors appreciate your comment and had added dates for data collection, accordingly.

Line 123- check spelling + the fact that some wore the device continuously might be better suited in results.

The authors appreciate your comment and had corrected the spelling mistake. Regarding the suggestion to move that some participants wore the device continuously, we decided to keep it in the methods section with other information about accelerometer-based analysis.

Line 125- what hours were labelled as sleep and was there any evidence used to inform this?

The authors appreciate your comment. Participants were asked to remove the device for water-based activities and during sleep time. As mentioned in line 125, we used Choi’s algorithm (2011) to calculate wear and non-wear time. As mentioned above (line 128) participants were also asked to use an activity diary to record on/off times.

Line 132- were there any measures used to identify non-wear time?

The authors thank you for your comment. As we informed in the previous comment, we had used Choi’s algorithm (2011) to calculate wear and non-wear time.

Line 138- could the authors provide some information about PA data distribution.

The authors appreciate your comment. PA data distribution had been shown in Table 1 using LPA and MVPA intensities for the description overall, and by countries; as well as in Table 2 for correlations with PCS and MCS; and Table 3 in the regression analysis.

Line 155- are the authors reporting on secondary data analyses- this is unclear

The authors are not reporting on secondary analysis. We included all the participants that fully completed the SF-12, the SBQ, the modified IPAQ and returned valid accelerometer data from the hip-worn ActiGraph.

Line 162- could the authors provide obesity results by sex

The authors appreciate your comment and had added obesity results by sex (Line 166).

Line 182- a percentage breakdown of participants by country would be informative. Perhaps another table?

The authors appreciate your comment and had added information by country in Table 1.

Line 325/326 this is an expected finding and previously reported. Do the authors believe we still need to further investigate this cross -sectionally or what do they advise?

The authors appreciate your comment and had deleted this first sentence of the ‘strengths and limitations’ section.

Round 2

Reviewer 1 Report

The authors have made significant improvements to the manuscript, but it would still require substantial revisions before publishing. My main concerns are:

1) The authors have now reported hours/day for accelerometer-based physical activity in Table 1, but it seems that the multivariate regression analyses have not been performed with these variables, as the values in Table 3 are identical to the original manuscript draft utilizing percentages of wear time.

2) The authors have now removed covariates from the Table 3, but as the values are identical to the original draft, my concern about the treatment of categorical covariates as continuous variables remains.

3) Introduction and discussion have been improved, but the rationale is still weak and the importance of the present study is not clear to the reader. The authors should justify better, why HRQoL is an important outcome.

Author Response

The authors have made significant improvements to the manuscript, but it would still require substantial revisions before publishing.

The authors thank the reviewer for his/her feedback. We have gone through the analysis again with an additional statistician and, as the reviewer suggested, there was a mistake that were not noticed by the authors in our final check. We appreciate the diligence of the reviewer in noticing it. We hope it is solved now.

My main concerns are:

1) The authors have now reported hours/day for accelerometer-based physical activity in Table 1, but it seems that the multivariate regression analyses have not been performed with these variables, as the values in Table 3 are identical to the original manuscript draft utilizing percentages of wear time.

The authors appreciate your comment and had performed the multivariate regression analyses with hours/day variables for accelerometer-based physical activity, as previously was calculated based on the percentage in LPA and MVPA.

2) The authors have now removed covariates from the Table 3, but as the values are identical to the original draft, my concern about the treatment of categorical covariates as continuous variables remains.

The authors appreciate your comment and had modified the multivariate regression analysis. We had treated all categorical covariates included in the model as categorical: country (taking as a reference Denmark), sex (taking as a reference women), BMI categories (taking as a reference obesity) and educational background (taking as a reference (university). We have added again all covariates’ values in Table 3 (models c and d).

3) Introduction and discussion have been improved, but the rationale is still weak and the importance of the present study is not clear to the reader. The authors should justify better, why HRQoL is an important outcome.

The authors appreciate your comment and have tried to highlight the importance of physical activity and sedentary behaviour in improving HRQoL. Specifically, we have tried to shift the focus to HRQoL in the introduction. Regarding the discussion, all the studies we mentioned based their goals to analyzing this variable and we have tried to better highlight it.

We hope that you will find our review suitable.
